# Hospital-based evidence on cost-effectiveness of brucellosis diagnostic tests and treatment in Kenyan hospitals

**Lorren Alumasa[1], Lian F. Thomas[1,2], Fredrick Amanya[1], Samuel M. Njoroge[1,3], Ignacio Moriyón[4], Josiah Makhandia[1], Jonathan Rushton[2], Eric M. Fèvre[1,2]\*, Laura C. Falzon[1,2]\***

**1** International Livestock Research Institute, Nairobi, Kenya, **2** Institute of Infection, Veterinary and Ecological Sciences, University of Liverpool, Liverpool, United Kingdom, **3** Kenya Medical Research Institute, Nairobi, Kenya, **4** Departamento de Microbiología y Parasitología, Facultad de Medicina, and Institute for Tropical Health, Universidad de Navarra, Pamplona, Spain

\* eric.fevre@liverpool.ac.uk (EMF); laura.falzon@liverpool.ac.uk (LCF)

**Data Availability Statement:** All data and the cost-effectiveness model have been uploaded to the

## Abstract

Hospitals in Kenya continue to use the Febrile Antigen *Brucella* Agglutination Test (FBAT) to diagnose brucellosis, despite reports showing its inadequacy. This study generated hospital-based evidence on the performance and cost-effectiveness of the FBAT, compared to the Rose Bengal Test (RBT).Twelve hospitals in western Kenya stored patient serum samples that were tested for brucellosis using the FBAT, and these were later re-tested using the RBT. Data on the running time and cost of the FBAT, and the treatment prescribed for brucellosis, were collected. The cost-effectiveness of the two tests, defined as the cost in US Dollars ($) per Disability Adjusted Life Year (DALY) averted, was determined, and a basic sensitivity analysis was run to identify the most influential parameters. Over a 6-month period, 180 patient serum samples that were tested with FBAT at the hospitals were later re-tested with RBT at the field laboratory. Of these 24 (13.3%) and 3 (1.7%) tested positive with FBAT and RBT, respectively. The agreement between the FBAT and RBT was slight (Kappa = 0.12). Treatment prescribed following FBAT positivity varied between hospitals, and only one hospital prescribed a standardized therapy regimen. The mean $/DALY averted when using the FBAT and RBT were $2,065 (95% CI $481-$6,736) and $304 (95% CI $126-$604), respectively. Brucellosis prevalence was the most influential parameter in the cost-effectiveness of both tests. Extrapolation to the national level suggested that an estimated $338,891 (95% CI $47,000-$1,149,000) per year is currently spent unnecessarily treating those falsely testing positive by FBAT. These findings highlight the potential for misdiagnosis using the FBAT. Furthermore, the RBT is cost-effective, and could be considered as the mainstay screening test for human brucellosis in this setting. Lastly, the treatment regimens must be harmonized to ensure the appropriate use of antibiotics for treatment.

University of Liverpool repository and are now available at the following link: https://doi.org/10.17638/datacat.liverpool.ac.uk/1200.

**Funding:** This work was supported by the Biotechnology and Biological Sciences Research Council, the Department for International Development, the Economic & Social Research Council, the Medical Research Council, the Natural Environment Research Council and the Defence Science & Technology Laboratory, under the Zoonoses and Emerging Livestock Systems (ZELS) programme, grant reference BB/L019019/1 [EMF]. It also received support from the CGIAR Research Program on Agriculture for Nutrition and Health (A4NH), led by the International Food Policy Research Institute (IFPRI). We also acknowledge the CGIAR Fund Donors (http://www.cgiar.org/funders) [EMF]. LFT is supported by the University of Liverpool-Wellcome Trust Institutional Strategic Support Fund and is a Soulsby Foundation One Health Fellow [LFT]. Research at the University of Navarra is supported by MINECO (PID2019-107601RA-C32) and the Institute for Tropical Health funders (Obra Social la CAIXA -LCF/PR/PR13/11080005 - and Fundación Caja Navarra, Fundación María Francisca de Roviralta, Ubesol and Inversiones Garcilaso de la Vega S.L) (https://www.unav.edu/web/instituto-de-salud-tropical/amigos) [IM]. The funders had no role in study design, data collection and analysis, decision to publish, or preparation of the manuscript.

**Competing interests:** The authors have declared that no competing interests exist

## Author summary

Brucellosis is the most common bacterial zoonosis globally, with a higher burden in low-resource settings. In humans, the disease manifests itself with non-specific clinical signs, and current international guidelines recommend the use of two serological diagnostic tests to make a confirmatory diagnosis. Many hospitals in Kenya and some neighbouring countries have been using the Febrile Antigen *Brucella* Agglutination Test (FBAT) for diagnosis, despite reports showing its poor performance. In this study we compared the diagnostic performance and cost-effectiveness of the FBAT with that of the Rose Bengal Test (RBT), a serological assay recommended by international guidelines. Our results showed that, compared to the RBT, the FBAT incorrectly diagnosed a number of patients. This is of concern as it leads to unnecessary antibiotic treatments, increasing the economic burden of the disease and exacerbating the risk of antibiotic resistance. We also highlight the discrepancies in brucellosis treatment regimens currently being prescribed by various hospitals. Finally, we showed that the RBT is a more cost-effective diagnostic test. Our recommendation, therefore, is for the RBT to be considered as the mainstay diagnostic test for human brucellosis in all Kenyan hospitals, and for the harmonization of treatment guidelines.

## Introduction

Brucellosis is a neglected zoonotic disease caused by Gram-negative, facultative intracellular bacteria belonging to the genus *Brucella* [1–5]. To date, six *Brucella* species have confirmed zoonotic potential [6], and those most frequently implicated in human brucellosis are *B. melitensis*, *B. abortus*, and *B. suis* [1,2,7].

Human brucellosis manifests itself as a febrile disease with a tendency towards long evolution and persistence. Other commonly reported clinical signs and symptoms include sweating, chills, general malaise, fatigue, headache and joint pains, though complications involving other organs and systems may also occur [1,2,8]. The lack of pathognomonic signs renders brucellosis clinically indistinguishable from other acute febrile diseases, and laboratory diagnostic tests are required for differential diagnosis and to ensure rational treatment.

While brucellosis has been eliminated in a number of developed countries, the disease persists endemically in many areas of the world. These include the Mediterranean basin, Middle East countries, Asia, South America, and North and East African countries, including Kenya [7,9–12], where our study is located. Several recent studies in markedly different ecological settings have shed light on the endemicity, though at different frequency levels, of human brucellosis in Kenya [11,13–18]. Indeed, while a national seroprevalence of 3% was reported [19], this may range between 1.0 to 2.4% in low-risk, smallholder tropical production systems areas such as Busia and Kiambu [13,16–17], and between 13.7 to 46.5% in higher risk, primarily pastoral areas such as Kajiado, Garissa, Tana River, Wajir and Marsabit [11,15–18]. Brucellosis was in fact ranked among the top five zoonotic diseases for prioritization in Kenya due to its high socio-economic impact and high prevalence in many Kenyan regions [20], and a recently developed National Brucellosis Strategy highlights the need for standardized diagnostic testing and treatment of brucellosis in Kenya [12].

Diagnostic testing, in both animals and humans, may rely on bacteriological, molecular, or immunological techniques. Bacteriological isolation and identification of the pathogen is considered the gold standard given its 100% specificity [1,21]. However, it is laborious and time-

consuming, it can only be performed in facilities with adequate biosecurity, and its sensitivity is diminished in focal or long evolution infections [2,5,7,22,23].

On the other hand, molecular techniques revolving around nucleic acid detection and amplification methods have a quicker turnaround time, and some have a higher sensitivity compared to bacterial culture methods [5]. However, there are no standardized protocols to ensure reproducibility among different laboratories, they cannot distinguish recovery from clinical disease, and require specialized equipment, limiting their use in low-resource settings [1,2,5,23].

Current serological assays detect the patient's antibody response towards the bacterial pathogen through agglutination, complement activation, immuno-precipitation reactions or in primary binding assays such as ELISA or lateral flow immunochromatography assays [1,21]. All these tests detect antibodies against the lipopolysaccharide antigen present on the outer membrane of smooth *Brucella* species, such as *B. melitensis*, *B. abortus*, and *B. suis* [5,21]. Thus, they could also detect antibodies to cross-reacting Gram-negative organisms, such as *Yersinia enterocolitica* O:9, *Vibrio cholerae*, *Francisella tularensis*, and *Escherichia coli* O116 and O157, resulting in false positives. Despite this, their simplicity and associated low cost make serological assays indispensable and affordable diagnostic tools, particularly in developing countries.

The Febrile Brucellin Antigen Test (FBAT) is a variant of the rapid slide *Brucella* agglutination test and is commonly used in East African countries given its low cost and simplicity [24,25]. However, studies have shown that it is misleading as it tends to underestimate the true positive cases while overestimating the overall prevalence [13,24]. The Rose Bengal Test (RBT) is also a rapid slide agglutination test performed at pH 3.7. However it has been shown to have a high diagnostic sensitivity and can detect *Brucella* specific antigenic stimulus in endemic settings. Moreover, it is relatively inexpensive, and requires the same basic laboratory equipment and expertise as the FBAT, making it optimal for small laboratories with limited means [13,22,23].

During ongoing surveillance activities in western Kenya [26,27], we observed that several hospitals were using the FBAT to diagnose human brucellosis. Furthermore, we noted that the hospitals prescribed different antibiotic regimes in case of a FBAT positive result. Brucellosis treatment is challenging given the intracellular nature of the pathogen, with frequent treatment failures or relapses [1,28,29]. In uncomplicated cases of brucellosis, the WHO recommends combinations of doxycycline (100mg twice daily for 45 days) and streptomycin (1g once daily for 14–21 days), or doxycycline (100mg twice daily for 45 days) and rifampicin (600-900mg once daily for 45 days) [9].

Driven by demand from our stakeholders, who are increasingly aware of the current discrepancies in brucellosis testing and treatment, this study was conducted to create hospital evidence on the cost-effectiveness of current brucellosis diagnostic tests and treatment regimes. Specifically, the study objectives were: (i) to determine the agreement between FBAT and RBT results; (ii) to document which FBAT kits are being used and the antibiotic treatment regimens prescribed; and (iii) to determine the cost-effectiveness of the FBAT compared to the RBT. The rationale was that self-generated hospital-level data could then support decision-making regarding diagnosis and treatment of brucellosis in Kenya.

## Materials and methods

### Ethics statement

This study was approved by the Institutional Research Ethics Committee (IREC Reference No. 2017–08) at the International Livestock Research Institute, a review body approved by the Kenyan National Commission for Science, Technology and Innovation. Approval to conduct

this work was also obtained from the Ministry of Health, the relevant offices at devolved government level, and the staff at each hospital.

## Hospital selection

Twelve hospitals participating in an integrated surveillance program for zoonotic diseases in the counties of Busia, Bungoma, and Kakamega in western Kenya [27] were included in this research activity. These twelve hospitals included three public County hospitals (one per County), three private Missionary hospitals (one per County), and six public sub-County hospitals (two per County), and comprised 33–44% of the hospitals in each County. To ensure that they were representative of all hospitals in the area, they were selected based on their catchment area and number of out-patients attending each site. Logistical factors (i.e. distance from central Busia laboratory) were also taken into consideration.

## Data and sample collection

At each hospital, we liaised with clinical staff or laboratory technologists who agreed to maintain records of a unique identifier assigned to each patient tested for brucellosis using the FBAT at their facility, the FBAT result, and any other diagnostic tests performed for that patient. Individuals were classified as positive if there was an agglutination reaction with any of the antigens in the diagnostic kit. Brucellosis diagnosis is normally considered when other conditions endemic to the area and presenting with similar symptoms, including non-specific acute or insidious onset of fever, are ruled out, or when patients were diagnosed and treated without improvement.

At each facility we requested that the anonymised serum samples that were tested for brucellosis with FBAT and labelled with the unique identifier were stored. These samples were preserved between 2–8˚C until our next field visit, when they were collected by one of our field team officers and transported in a cool box with two icepacks to our field laboratory in Busia for further testing with the RBT.

At each hospital, we also collected information on the FBAT performed and the treatment prescribed to those patients that test positive. Specifically, the clinical officers and laboratory technicians provided us with information on the test kit manufacturer, the total fee charged to the patient for testing using the FBAT, how they carry out the test, and how long it takes to execute one FBAT. When available, we made a copy of the test instruction sheet provided with the diagnostic kit. For those hospitals which had started to run the RBT, we also obtained the price they charged patients for this test. Treatment information, including the name of the drugs, dosage, and treatment duration prescribed, and the cost charged to the patient for such treatment, was obtained from the clinical officers and pharmacists. All information was collected at hospital-level, and no individual data on the actual treatment prescribed to each patient was obtained.

## Laboratory analysis

Serum samples brought to the field laboratory in Busia were preserved at 2–8˚C and re-tested with the RBT within five days. Single blinding was performed, whereby those performing the RBT were not aware of the FBAT results to avoid misclassification bias.

The diagnostic antigen (prepared and controlled for quality following established guidelines [30]) was supplied by the Instituto de Salud Tropical Universidad de Navarra, in Pamplona, Spain, and the test was carried out as follows. Both sera samples and antigen were brought to room temperature. An automatic pipette was used to dispense 25µl of the sample on to the glossy side of a white tile, and an equal volume of the antigen was dispensed next to each drop

of serum. The antigen and each serum sample were mixed immediately using individual, non-treated wooden splints, and the plate rocked gently for four minutes. Following this, the results were read immediately in a well-lit place and interpreted as either positive or negative. Samples were considered positive when any degree of visible agglutination was observed, as previously described [22].

## Data capture and analysis

Data from the hospital records with the FBAT results, and the lab records with the RBT results, were entered manually into an Excel spreadsheet (Microsoft, Redmond, WA, USA), while data cleaning and analysis were carried out using Stata Statistical Software: Release 14 (College Station, TX: StataCorp LP) and @Risk 7.5 (Palisade, Newfield, NY, USA) add-on for Microsoft Excel.

**FBAT and RBT results.** The proportion of patients that tested positive for brucellosis based on the FBAT performed at the hospital, and the RBT performed at the field laboratory, was determined. This was then used to determine the inter-test Kappa agreement between the FBAT and RBT results, and the Kappa agreement score was interpreted using the scale described by Dohoo et al. [31]. A McNemar's Chi$^2$ test was computed to determine whether the contingency table for the compared tests was symmetrically distributed, whereby a $p$-value <0.05 was considered statistically significant and indicative that the contingency table was asymmetrically distributed.

Details of the FBAT kit manufacturers, how the test is carried out at each hospital, the test running time, and the cost charged to the patient, together with the hospital-level treatment regimens prescribed and their cost, were summarized. Available test instruction sheets were also reviewed and summarized.

**Cost-effectiveness of FBAT vs. RBT in the study population.** A basic stochastic cost-effectiveness model from the societal perspective was built using @Risk 7.5 (Palisade, Newfield, NY, USA) add-on for Microsoft Excel. This model calculated the cost-effectiveness of the two tests, where costs and benefits are assumed to be experienced across the whole of society, regardless of who pays the actual costs, defined as the cost in US Dollars ($) per Disability Adjusted Life Year (DALY) averted over the course of one year. The model used the parameters and associated probability distributions outlined in Table 1, which were based on data from this study and published literature. Specifically, raw data for prevalence estimates and diagnostic test performance estimates were used to parameterise a Beta distribution using the epitools.ausvet calculator for estimating parameters for Beta distributions from count data (https://epitools.ausvet.com.au/betaparamsmultidata), and the distributions were truncated with a minimum of 0 and maximum of 1 using the RiskTruncate(0,1) function in @Risk. Data on the diagnostic performance of the FBAT were based on a study by Kiambi et al. [32], which compared the performance of a FBAT (Febrile Serodiagnostics, Biosystems, Spain) performed qualitatively (i.e. by mixing 50 μl of serum with a drop of the rapid test reagent and observing for an agglutination reaction within two minutes) with PCR.

Data on the time taken to run each diagnostic test, the cost of the test, and the cost of treatment from this study were used directly to fit a distribution in @Risk using the 'distribution fitting' function which fits the available data to different distributions. The distribution with the lowest corrected AIC value (AICc which includes a correction for small sample size) was then chosen for use in the model. Where only a minimum to maximum range was available, a uniform distribution was utilised to account for more uncertainty in the estimate.

The different parameters and scenarios for the comparative cost-efficacy analysis are shown in Table 1. The primary outcome of interest was the cost-effectiveness of the two tests in our

**Table 1. Parameters used for the comparative cost-effectiveness analysis of the Febrile Antigen *Brucella* Agglutination Test and the Rose Bengal Test.** Parameters used for all scenarios unless otherwise indicated.

| | Parameter | Distribution | Data | Source |
|---|---|---|---|---|
| **P1** | Number of patients tested/year | Uniform | ***Scenario 1*** <br> 488 (439–537) patients presented for testing in 1 year across 10 hospitals | Extrapolated from current study (285 patients reported for testing during the 7- month follow-up) |
| | | Static | ***Scenario 3*** <br> 77,873 | [33] <br> Brucellosis cases reported to DHIS from across the country in 2012 |
| **P2** | P(brucellosis) | Beta(9,818) | ***Scenario 1*** <br> 0.01 (95% CI 0.004–0.02) <br> 8 cases of 825 tested | [13] <br> *Brucella* spp. prevalence in febrile patients presenting to hospital in western Kenya based on qualitative RBT |
| | | Beta(61,327) | ***Scenario 2*** <br> 0.154 (95% CI 0.12–0.195) <br> 60 cases in 386 tested | [32] <br> *Brucella* spp. prevalence in febrile patients presenting to hospital in North-East Kenya based on real-time PCR |
| | | Beta (34,1068) | ***Scenario 3*** <br> 0.03 (95% CI 0.01–0.05) <br> 33 cases from 1091 samples tested | [19] <br> National seroprevalence of brucellosis based upon 1091 serum samples obtained through the 2007 Kenya AIDS Indicator Survey |
| **P3** | P(not brucellosis) | | 1-P2 | |
| **P4** | P(true positive FBAT) | Beta (23,39) | 0.37 (0.25–0.50) <br> 22/60 cases detected by FBAT | [32] |
| **P5** | P(false negative FBAT) | | 1-P4 | |
| **P6** | P(true negative FBAT) | Beta(227,101) | 0.69 (0.64–0.74) <br> 226/326 non-Brucella cases correctly identified by FBAT | |
| **P7** | P(false positive FBAT) | | 1-P6 | |
| **P8** | P(true positive RBT) | Beta(254,1) | 253/253 confirmed Brucella cases detected by RBT | [22,34] |
| **P9** | P(false negative RBT) | | 1-P8 | |
| **P10** | P(True negative RBT) | Beta(1646,5) | 0.97 (0.92–0.99) | |
| **P11** | P(false positive RBT) | | 1-P10 | |
| **P12** | FBAT cost to patient | RiskPareto(5.5183,1.5) | Table 4 | Current study, fit using 'distribution fitting' in @Risk |
| **P13** | RBT cost | Uniform | $1.50-$3.26 | Current study |
| **P14** | Antibiotic treatment costs | RiskExpon(12.837, RiskShift(1.6953)) | Table 4 | Current study, fit using 'distribution fitting' in @Risk |
| **P15** | Disability weighting brucellosis | Uniform | 0.15–0.211 | [8] |
| **P16** | Mean duration of untreated brucellosis | RiskExpon(0.45, RiskShift(4.5)) | 4.5 years | [35] |
| **P17** | Years of Life lived with Disability (YLD) brucellosis | | = *Number of cases (P1 x P2) x disability weighting (P15) x duration of illness (P16)* | |
| **P18** | Years Life Lost (YLL) brucellosis | Uniform | 0–0 | [8] <br> where no mortality, excluding abortions, was reported |

(*Continued*)

**Table 1.** (Continued)

| Parameter | Distribution | Data | Source |
|---|---|---|---|
| **P19** DALYs attributed to untreated case of brucellosis | | *= Years Lost due to Disability (P17) + Years of Life Lost (P18)* | |

P(x) = Probability(event); DALY = Disability Adjusted Life Years

Cost-effectiveness ($/DALY averted within the tested population) for each test was calculated as follows

= Total cost of testing all patients and treating true positive & false positive patients in a year / DALYs averted in a year by treating true positive cases

Where

*Total cost testing all individuals = number patients tested x cost of diagnostic testing*

*True positives = Number screened (P1) x prevalence x test sensitivity*

*False positives = Number screened (P1) x (1 –prevalence) x (1 –test specificity)*

*DALYS averted = number of true positives treated per year x DALYS attributable to an untreated case (P19)*

study site in western Kenya *(Scenario 1)*. We next explicitly illustrated the influence of brucellosis prevalence on the cost-effectiveness of the tests by running a further analysis for a high-prevalence situation. For this, we used a recently published prevalence of brucellosis in Ijara district hospital in North-East Kenya of 15.4% (95% CI 12–19.5%) [32], with the assumption that all other parameters remained the same *(Scenario 2)*. Finally, to generalise our findings and give an understanding of the impact a change of diagnostic policy would have at the national level, a separate analysis was run to extrapolate our study results to the national level *(Scenario 3)*. In this scenario we used the number of brucellosis cases reported in the District Health Information System (DHIS) [33], and the estimated national prevalence [19]. The model structure is illustrated in Fig 1, while the Microsoft Excel spreadsheet with probability distributions and all relevant datasets can be accessed through this link: https://doi.org/10.17638/datacat.liverpool.ac.uk/1200

Several assumptions were made in this model: 1) The cost charged to the patient by the hospital, which varied by hospitals and by throughput in different hospitals, included full cost-recovery for the hospital, including consumables and staff time for sample collection, processing, and testing. 2) The opportunity cost to the patient of the two tests was assumed to be equivalent and therefore not included in this analysis. 3) The cost of treatment assumed that each patient who tested positive for brucellosis underwent one full treatment protocol, and that there was 100% compliance with the treatment. This is likely to be an over-estimation, yet without data on compliance levels we felt it was more appropriate to retain this value for both testing scenarios, therefore also assuming that there was no change in compliance based on testing regime. 4) Patients were assumed to seek treatment at the onset of brucellosis-like symptoms, and the DALYs averted for treating a case of brucellosis were assumed to incorporate the full duration of the untreated disease, as suggested by Roth et al. [35]. 5) As an 'official' disability weighting has not been assigned to brucellosis through the global burden of disease study, the range of weightings suggested by Dean et al. [8] was used, with no differentiation of the different clinical manifestations presenting in this study since we did not have sufficient data for a more accurate calculation of the DALY burden in this population. We emphasise that we were not attempting an absolute estimate of DALYs but were simply using a calculated DALY outcome to estimate cost-effectiveness. 6) The potential DALY burden imposed upon the community through the inappropriate use of antimicrobial treatment of false positives (and subsequent resistance which may emerge) was also not quantified in this study. 7) No attempt was made to quantify under-reporting of brucellosis from patients who did not present to health care, and we assumed that this rate remained constant irrespective of the

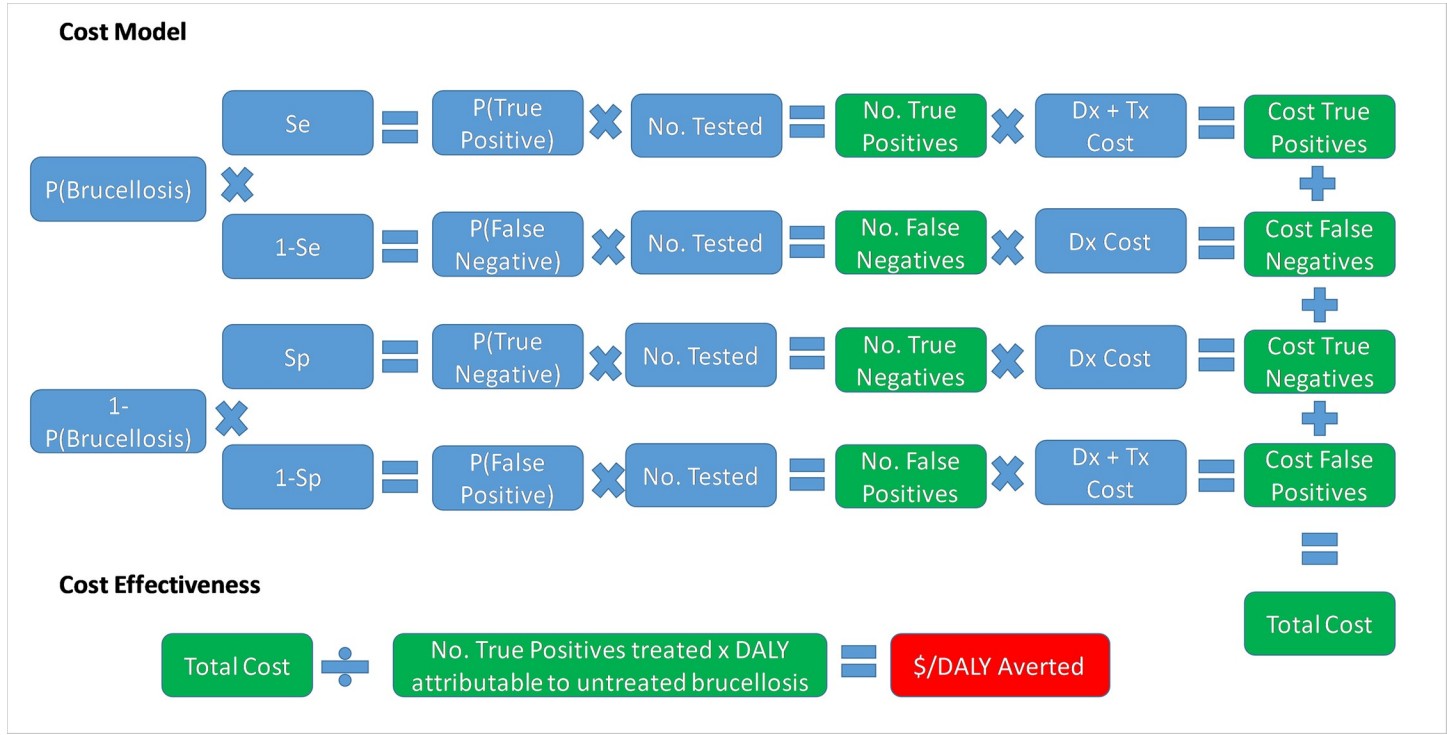

**Fig 1. The model structure used for the comparative cost-effectiveness analysis of the Febrile Antigen *Brucella* Agglutination Test and the Rose Bengal Test.**

diagnostic procedure used. 8) In the case of the national level extrapolation, we assumed that the reported national prevalence is equivalent to the prevalence in a population seeking treatment for febrile illness and suspected to have brucellosis based on clinical judgement; there is therefore substantial uncertainty around the national-level estimates and they should be taken for illustrative purposes only.

The 'Auto' function in @Risk was used to run sufficient iterations of the model until all input parameters converged using the default values of 3% tolerance and 95% confidence. The influence of the input parameters on the outcome of interest were explored through the calculation of Spearman rank correlation coefficients (ρ values) in @Risk. The ρ value can range from -1 to +1 and indicate the strength of association between the input and output variable and the direction of the relationship.

## Results

### Comparison of FBAT and RBT results

Ten hospitals participated in this study between October 2018 and April 2019; these included three County Hospitals, three Missionary hospitals, and four Sub-County hospitals.

In total, records for 284 patients who were tested for brucellosis at any one of these facilities were obtained. However, 104 were either missing an FBAT result (n = 79) or corresponding serum sample (n = 25) and were therefore discarded. Of the remaining 180 samples with complete patient records, 24 (13.3%) tested positive with the FBAT performed at the hospital, though the proportion of those testing positive for brucellosis with the FBAT varied considerably among the participating hospitals (Table 2).

When these samples were re-tested with the RBT at the field laboratory, 3 (1.7%) tested positive. Two of the three samples that tested positive with RBT (one in Hospital B and one in

**Table 2. Brucella test results obtained with the Febrile Antigen *Brucella* Agglutination Test (FBAT) performed at the hospital, and with the Rose Bengal Test (RBT) performed at the Busia field laboratory, for patients visiting 10 hospitals in western Kenya.**

| Hospital | FBAT kit | Antigens | No. samples tested | No. FBAT positive | % FBAT positive | No. RBT positive | % RBT positive |
|---|---|---|---|---|---|---|---|
| A | N/A[1] | N/A | 4 | 2 | 50.0 | 0 | 0 |
| B | OZOTEX *Brucella* Agglutination Slide test [medSource] | *Brucella abortus Brucella melitensis* | 67 | 10 | 14.9 | 2 | 3.0 |
| C | Febrile Kit *Brucella abortus/melitensis* [SeroLab] | *Brucella abortus Brucella melitensis* | 17 | 2 | 11.8 | 0 | 0 |
| D | OZOTEX *Brucella* Agglutination Slide test [medSource] | *Brucella abortus Brucella melitensis* | 16 | 3 | 18.8 | 0 | 0 |
| E | N/A | N/A | 4 | 0 | 0 | 0 | 0 |
| F | Antigens procured from HR Drugs/631 B(H) | *Brucella abortus Brucella melitensis* | 13 | 0 | 0 | 0 | 0 |
| G | Expert Febrile Antigens Slide/Tube Test [Expert Diagnostics] | *Brucella Proteus Salmonella* O *Salmonella* H | 7 | 0 | 0 | 0 | 0 |
| H | Febrile Kit *Brucella abortus/melitensis* [Fortress diagnostics] | *Brucella abortus Brucella melitensis* | 4 | 0 | 0 | 0 | 0 |
| I | Diagnostic kit for determination of *Brucella abortus/ melitensis* antibodies [Accurate] | *Brucella abortus Brucella melitensis* | 7 | 5 | 71.4 | 0 | 0 |
| J | Expert Febrile Antigens Slide/Tube Test [Expert Diagnostics] | *Brucella Proteus Salmonella* O *Salmonella* H | 41 | 2 | 4.9 | 1 | 2.4 |
| **Total** | | | **180** | **24** | **13.3** | **3** | **1.7** |

[1]N/A, not available

Hospital J) had also tested positive with the FBAT performed at the hospital, while the other RBT-positive sample in Hospital B had tested negative with the FBAT. Twenty-two samples that tested positive with the FBAT performed at the hospital, tested negative with the RBT.

The Kappa agreement for the two tests was 0.12, indicating a slight agreement. Moreover, the McNemar's Chi$^2$ test was statistically significant (p-value<0.001), indicating that the contingency table for the two compared tests was not symmetrically distributed and therefore biased. Specifically, a bias was observed whereby the FBAT was more likely to classify a patient as positive for brucellosis, compared to the RBT.

Patients who were tested for brucellosis were regularly tested for other diseases. The most frequently conducted diagnostic tests included blood smears to test for malaria (n = 123), tests for rheumatoid factor (n = 48), the Standard Agglutination Test for *Salmonella typhi* O and H antibody titres (n = 27), and full blood haemograms (n = 22).

### Details on diagnostic test kits and prescribed treatments

Details on the FBAT kit and antigens used in the hospitals are presented in Table 2, while the test running time and cost, together with the treatment prescribed to the patients who tested positive for brucellosis, are presented in Table 3.

At the time of the study, all participating hospitals were conducting the FBAT as a routine screening test for brucellosis in febrile patients. The diagnostic kits used in these seven hospitals were sourced from five different manufacturing companies (SeroLab and Fortress had the same postal address on the test instruction sheet) (Table 2). All kits except one included two separate antigens for *B. abortus* and *B. melitensis*. In all available test instruction kits, the recommendations were to run a qualitative rapid slide test, and then confirm positive results with a semi-quantitative slide titre test or quantitative tube agglutination test. However, all hospitals only reported carrying out the qualitative rapid slide test, which corroborates with the reported mean FBAT running time of 35 minutes (ranging between 15 and 60 minutes). Only one hospital (hospital J) reported that they occasionally conduct titrations on serum samples of patients who visit their facility to re-test for brucellosis after having been diagnosed and treated for brucellosis in other facilities. In this case they conduct titrations to assess for possible increases or decreases in titres. The publication dates of the references cited in the FBAT

**Table 3. The running time and cost of the Febrile Antigen *Brucella* Agglutination Test (FBAT), the treatment prescribed to patients who tested positive for brucellosis and its cost in hospitals in western Kenya.**

| Hospital | Hospital type | Test running time (mins) | Test cost (US $) | Treatment prescribed | Consistent with WHO guidelines | Total treatment cost (US $) |
|---|---|---|---|---|---|---|
| A | County hospital | N/A[1] | N/A | N/A | N/A | N/A |
| B | Sub-County hospital | 30 | 1.50 | Doxycycline 100 mg bd[2] for 21 days plus Cotrimoxazole 960 mg bd for 21 days | Partly (correct drug combination & dosage but incorrect duration) | 3.30 |
| C | Missionary hospital | 15 | 2.00 | Doxycycline 100 mg bd for 21 days plus injectable Ceftriaxone 1g od[3] for 5 days | No | 33.60 |
| D | County hospital | 20 | 3.00 | Doxycycline 100 mg bd for 45 days plus Rifampicin 150mg bd for 45 days | Yes | 34.90 |
| E | Sub-County hospital | N/A | N/A | N/A | N/A | N/A |
| F | Missionary hospital | 60 | 2.00 | Amoxicillin trihydrate-potassium clavulanate 625 mg for 14 days | No | 13.60 |
|  |  |  |  | OR |  |  |
|  |  |  |  | Cefuroxime axetil 500 mg for 14 days | No | 13.60 |
| G | Sub-County hospital | 40 | 1.50 | Doxycycline 100 mg bd for 21 days plus Cefuroxime 500 mg bd for 14 days | No | 6.80 |
|  |  |  |  | OR |  |  |
|  |  |  |  | Doxycycline 100 mg bd for 21 days plus Amoxiclav 625 mg bd for 10 days | No | 7.80 |
| H | Missionary hospital | 20 | 1.50 | Doxycycline 100 mg bd for 21 days plus Co-trimoxazole 960 mg bd for 21 days | Partly (correct drug combination & dosage but incorrect duration) | 15.50 |
| I | Sub-County hospital | 60 | 1.50 | N/A | N/A | N/A |
| J | County hospital | N/A | N/A | N/A | N/A | N/A |

[1] N/A, not available

[2] bd, twice daily

[3] od, once daily

protocols ranged between 1916 and 1989. The mean fee charged to the patient for a FBAT was $1.80 (ranging between $1.50 and $3.00).

Two of the participating hospitals (A and B) started to run the RBT shortly after completion of data collection. The costs for testing provided by these two hospitals were $2 and $3 respectively. A cost of $3.26 was calculated as the cost of running the RBT within the Busia field laboratory where testing was ongoing for over 4000 human and animal samples as part of a larger study [27]. The costs associated with the collection and running of samples within this study included (1) technician time for sample collection, processing and running the test; (2) consumables including gloves, vacutainer tube and blood collection set, cotton wool, micropore tape, Eppendorf tube (1.5ml), 3ml transfer pipette, tooth pick, RBT fluid; and (3) biohazard disposal which included a portion of sharps container, autoclave bag, tape and the incineration fee. We assumed this as the upper limit of RBT costs (Table 1) due to the low purchasing power and use of an expensive blood collection set.

Clinicians in all participating hospitals reported that all patients who test positive with the FBAT receive prescriptions, though the treatment prescribed varied markedly between hospitals, both in terms of drug classes and duration. The prescribed treatment duration ranged between 14 and 45 days and cost $16.14 (ranging between $3.30 and $34.90). Only one of the six hospitals (hospital D) prescribed a standardized and recommended therapy regimen. Two other hospitals (hospitals B and H) prescribed accepted drug combinations at correct dosages, but the treatment duration for both doxycycline and rifampicin was too short (i.e. 21–30 days instead of 45 days). The three other hospitals prescribed drugs which are not mentioned in any recommendations (Table 3).

## Comparative cost-effectiveness of RBT vs FBAT in this study population

The model converged after 5400 iterations; key outputs are summarised in Table 4 and can be explored in detail in the model available at: https://doi.org/10.17638/datacat.liverpool.ac.uk/1200. In western Kenya (*Scenario 1*), the mean $/DALY averted across the studied population when using the FBAT was $2,065 (95% CI $481-$6,736), while the mean $/DALY averted across the studied population when using the RBT for diagnosis was $304 (95% CI $126-$604), indicating that a shift to RBT would be cost-effective, compared to the FBAT. The RBT remains cost-effective relative to the FBAT across each scenario modelled (Table 4 and Figs 2–4).

When considering the cost of treating false positives, the economic burden of 'unnecessary' treatments in the western Kenya study site is currently $2,166 (95% CI $297–7,359) per year, in comparison to the potential burden of unnecessary treatments of $21 (95% CI $2-$85) per year under a RBT diagnosis regime (Table 4). Extrapolating to the national level illustrates what a change in diagnostic policy may mean at this scale, demonstrating that potentially $338,891 (95% CI $47,000-$1,149,000) is currently spent on unnecessary treatments due to misdiagnosis, compared to a projected $3,344 (95% CI $317-$13,159) should the country shift to the RBT (Table 4).

**Table 4. Summary results of the cost-effectiveness model comparing the Febrile Antigen *Brucella* Agglutination Test (FBAT) with the Rose Bengal Test (RBT) in the three scenarios modelled.**

| Scenario | Scenario 1 (Low prevalence setting—western Kenya) | | Scenario 2 (High prevalence setting—north-eastern Kenya) | | Scenario 3 (National extrapolation) | |
|---|---|---|---|---|---|---|
| Diagnostic Test | FBAT | RBT | FBAT | RBT | FBAT | RBT |
| **$/DALY averted** | $2,065 ($481-$6,736) | $304 ($126-$604) | $132 ($40 - $364) | $34 ($15-$76) | $667 ($194-$1,915) | $109 ($56-$184) |
| **$/year treating false positive** | $2,166 ($297-$7,359 | $21 ($2-$85) | $1,845 ($252 - $6,267) | $18 ($2-$72) | $338,891 ($47,000-$1,149,000) | $3,344 ($317-$13,159) |

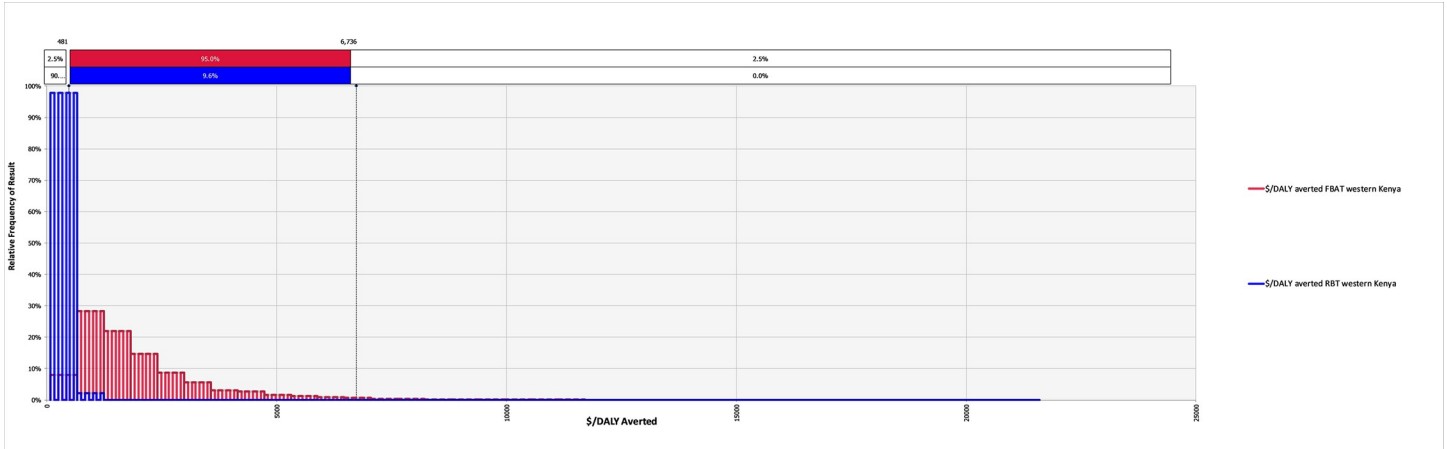

**Fig 2. The relative frequency of results for the cost-effectiveness, in terms of $/DALY averted, for the Febrile Antigen *Brucella* Agglutination Test and Rose Bengal Test in western Kenya *(Scenario 1)*.**

Spearman rank coefficients were calculated in @Risk to determine the influence of input parameters on the outcomes '$/DALY averted' for the FBAT and RBT tests under Scenario 1 (western Kenya). Ranking input parameters by their ρ values illustrated that the prevalence of brucellosis in the region and the cost of treatment were the most influential parameters influencing the cost-effectiveness of the FBAT (Fig 5). Specifically, the cost-effectiveness is improved with increasing prevalence as the positive predictive value improves and fewer false positives are treated. On the other hand, the cost-effectiveness is unsurprisingly negatively correlated with the cost of treatment. The underlying prevalence of brucellosis was by far the most influential factor in the cost-effectiveness of the RBT, again with higher prevalence being related to improved cost-effectiveness (Fig 6).

## Discussion

This study was done to generate hospital-based evidence on the cost-effectiveness of the RBT, relative to the FBAT, as a diagnostic tool for human brucellosis. We also shed light on the

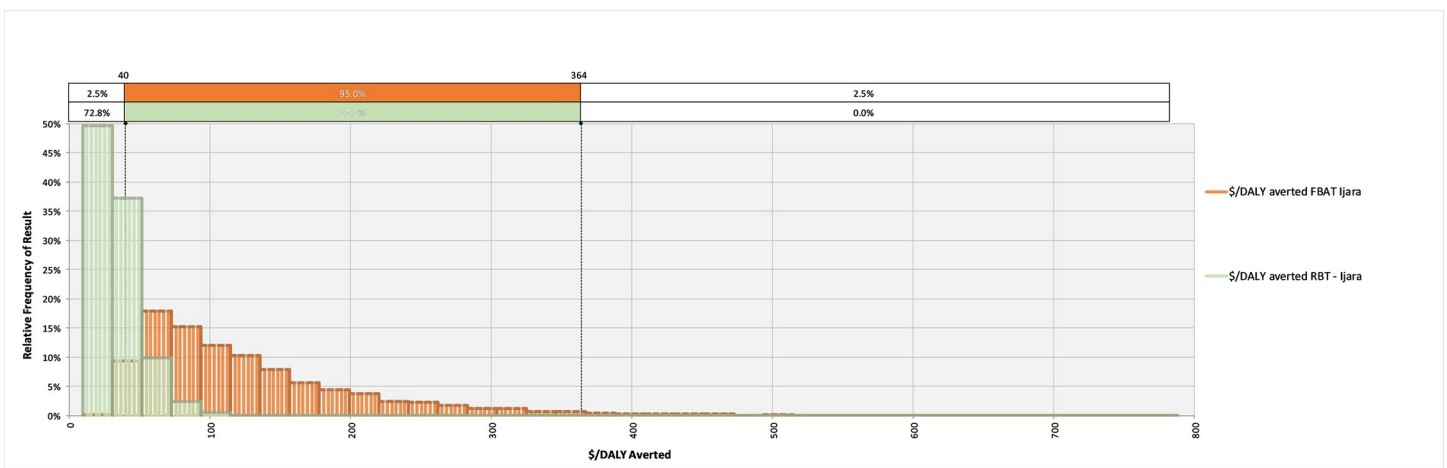

**Fig 3. The relative frequency of results for the cost-effectiveness, in terms of $/DALY averted, for the Febrile Antigen *Brucella* Agglutination Test and Rose Bengal Test in northern Kenya *(Scenario 2)*.**

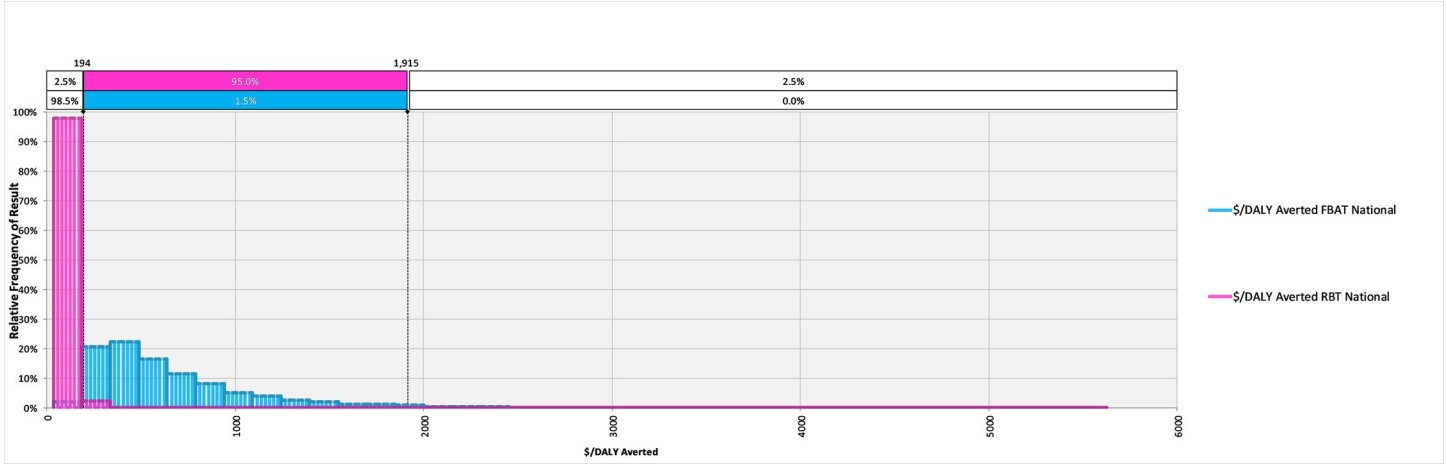

**Fig 4. The relative frequency of results for the cost-effectiveness, in terms of $/DALY averted, for the Febrile Antigen *Brucella* Agglutination Test and Rose Bengal Test nationally (*Scenario 3*).**

various FBAT kits being used, how the test is carried out, together with treatment regimens being prescribed at the participating hospitals. A national strategy for the prevention and control of brucellosis in Kenya is currently being discussed [12], and these study findings may aid discussions on the standardization of brucellosis diagnostic and treatment regimens in Kenya, and elsewhere where FBAT is still routinely used.

## Comparison of FBAT and RBT results

At the time of the study, all participating hospitals were using the FBAT to diagnose brucellosis, though since then, and as a result of our early analyses, a couple of hospitals (A and B) have switched to the RBT. The FBAT has long been discarded in Europe and the US, and was not even included among the outdated and obsolete tests in a recent review on the laboratory diagnosis of human brucellosis [5], yet it continues to be actively marketed and sold in health care settings in sub-Saharan Africa. Moreover, the claim that the test can differentiate between *B. melitensis* and *B. abortus* is misleading at best. The two bacterial species are antigenically similar and therefore cannot be differentiated by serological techniques [5,7,21].

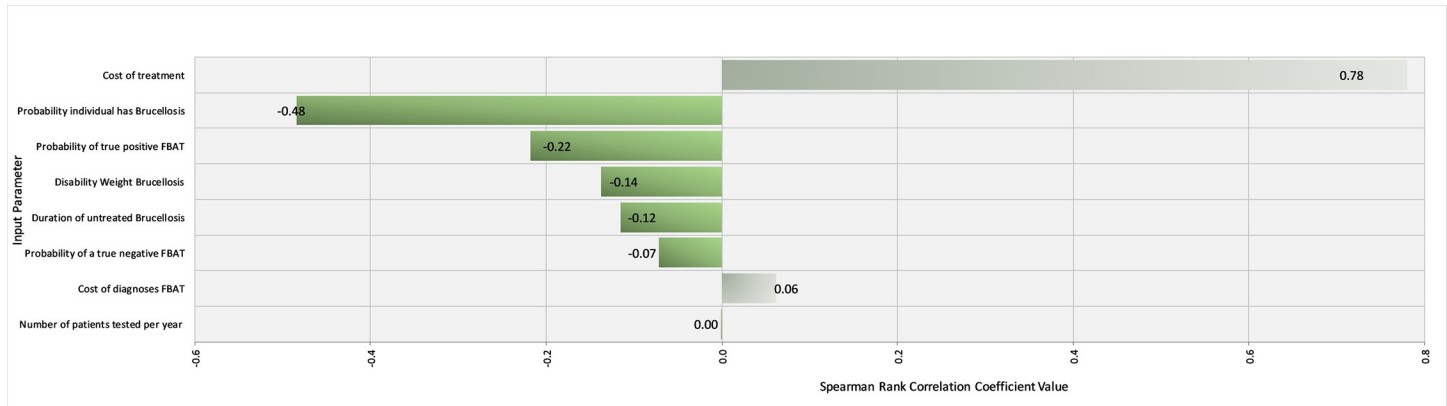

**Fig 5. Tornado Graph of Spearman Rank Correlation Coefficient values for different input parameters illustrating those most influencing the cost-effectiveness of the Febrile Antigen Brucella Agglutination Test in western Kenya.**

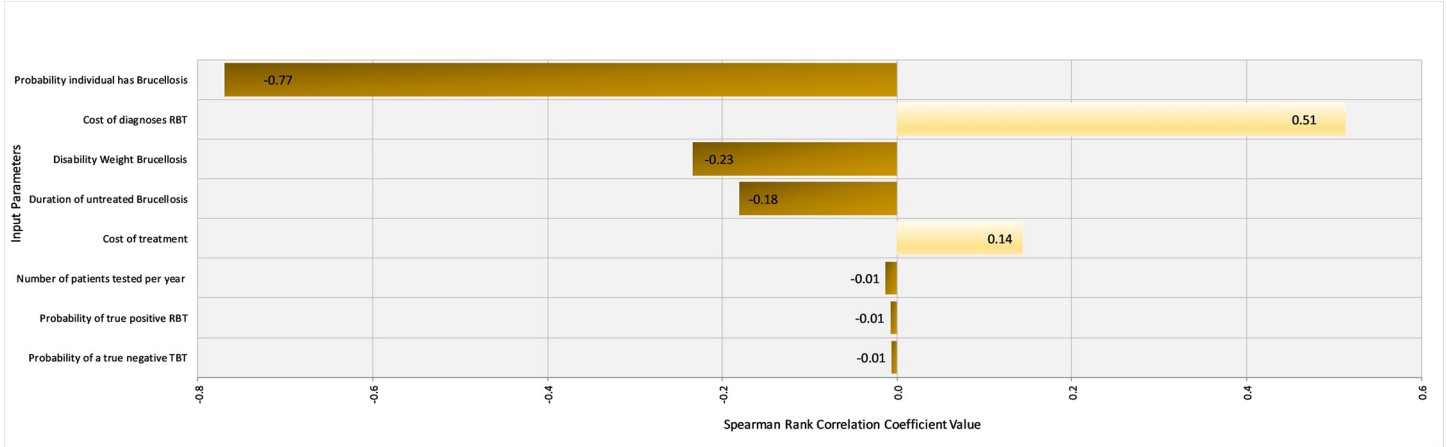

**Fig 6. Tornado Graph of Spearman Rank Correlation Coefficient values for different input parameters illustrating those most influencing the cost-effectiveness of the Rose Bengal Test in western Kenya.**

In this study, 13.3% of the serum samples tested with the FBAT at the hospitals were brucellosis-positive, compared to 1.7% of those re-tested with the RBT at the field laboratory. Unsurprisingly, there was substantial disagreement between the FBAT and RBT results, similar to previous findings which report that the diagnostic performance of the RBT is superior to that of the FBAT [13,32]. Of particular concern is the over-diagnosis of brucellosis when using the FBAT, as evidenced by the significant McNemar's Chi$^2$ test reported in this study. This test indicates that not only did the two tests disagree, but also that the nature of disagreement was biased, whereby the FBAT was more likely to give a positive outcome compared to the RBT. The high number of possibly false positive test results has a number of implications. Firstly, the misdiagnosis of brucellosis means that other clinically indistinguishable diseases may go unnoticed and, consequently, untreated. Secondly, the misdiagnosis of brucellosis leads to inappropriate and, in some cases unnecessary, antibiotic treatment. Lastly, misdiagnosis and mistreatment based on FBAT results also has important economic repercussions. Patients who are misdiagnosed often remain sick and therefore need to get re-tested and re-treated, as has been seen with other febrile issues such as trypanosomiasis [36], increasing the healthcare expenses associated with diagnostics and medications. Furthermore, the patient continues to suffer from poor health and emotional stress, and may be unable to work [1,37]. All these factors increase the societal impact and economic burden of the disease.

### Details on diagnostic tests and prescribed treatments

The FBAT kits used at the participating hospitals were sourced from different manufacturing companies, which might explain the observed variation in the proportion of those testing positive with FBAT among the different hospitals (Table 2). However, the sample size of those tested at each hospital is too small to allow for more meaningful comparisons between the tests.

We noted that the references listed in the test protocols were outdated, further suggesting that this is an obsolete technology. Furthermore, only the Accurate test kit (used in hospital I) reported a diagnostic sensitivity and specificity. However, there were no studies to substantiate these claims. We therefore opted to use FBAT diagnostic parameters published by Kiambi et al. [32] (Table 1) for our cost-effectiveness analysis, particularly since the FBAT in this study was carried out qualitatively, similar to the methodology in our participating hospitals.

In this study, only one hospital (hospital D) was compliant with the WHO treatment recommendations; two hospitals (hospitals B and H) were partly compliant, while three hospitals prescribed drugs which are not mentioned in any recommendations. Noteworthy was the fact that the same treatment was five times more expensive in hospital H than in hospital B. The reason for the discrepancy is that Hospital B is a government funded public health facility where costs are often subsidized to promote equity in access to services and individual treatment. Hospital F, on the other hand, is a private facility where costs are higher since patients can afford health insurance or out-of-pocket payment.

Studies have shown that beta-lactams, such as amoxicillin, do not have any *in vitro* activity towards *Brucellae* [38], while a study conducted by Lang et al. [39] showed that the use of ceftriaxome for treatment of human brucellosis resulted in high treatment failure and relapse. For these reasons, neither beta-lactams nor cephalosporins are warranted for the treatment of human brucellosis. Besides the potential for drug toxicity given the long treatment duration, this inappropriate use of drugs may also exacerbate the challenge imposed by drug resistance [1,29,38]. This is of particular concern since one of the antibiotics used in the treatment of brucellosis, rifampicin, is among the first-line medicines used for treatment of human tuberculosis. Rifampicin-resistant or multi-drug tuberculosis (MDR-TB) was declared a global emergency by the World Health Organization in 2014, and Kenya ranks among the thirty countries with highest burden of MDR-TB [40]. Furthermore, third-generation cephalosporins such as ceftriaxome, are the best available antibiotics for treating drug resistant bacteria, and should therefore be used judiciously [41].

It is unclear why many hospitals are not following the WHO or other recommendations regarding appropriate brucellosis treatment regimens. There is a lack of national guidelines for control; it is possible that there is inadequate information among healthcare providers, or clinicians may be prescribing shorter treatment durations (i.e. 21 vs. 45 days) to limit the expenses incurred by the patient. The use of alternative drugs, such as beta-lactams and cephalosporins, could also be an empiric way of managing other bacterial diseases which could be responsible for the patient's malaise. Indeed, one of the pharmacists participating in this study mentioned that some of the antibiotics prescribed, including doxycycline, ceftroxime and amoxicillin, still seemed to work on patients because the majority did not really have brucellosis but rather another bacterial infection (Alumasa, personal communication). This once again highlights the urgency to improve diagnostic methods and harmonize national guidance on optimal treatment regimens.

## Comparative cost-effectiveness of RBT vs. FBAT

The economic analysis demonstrates the superior cost-effectiveness of the RBT in terms of $/DALY averted within this study population. The cost-effectiveness of each diagnostic test is, unsurprisingly, improved when we investigate their use in a higher prevalence area, due to the superior positive predictive value in these situations. Nonetheless, the RBT still remains superior, both in terms of diagnostic sensitivity and specificity, and cost-effectiveness as measured by $/DALY averted across each scenario modelled. This despite the current higher cost of running the RBT, per sample, in our study hospitals. It is likely that the cost of operating the RBT would reduce with greater uptake in the country due to increasing economies of scale: for comparison, the total cost for running the RBT (including sample collection, sera separation and staff time) by a research orientated programme with a relatively small purchasing power is $3.26 [27]. Should large scale purchase of the RBT reagents and consumables (e.g. through government procurement) lead to a reduction in the cost of diagnosis, the relative cost-effectiveness of RBT will be further improved.

The analysis presented in this manuscript indicate that the FBAT, in both western Kenya and nationally, results in additional spending treating false positive cases. Specifically, the

direct economic losses due to 'unnecessary' treatments currently amount to approximately $2,166 per year across our study site (95% CI $297-$7,359). When extrapolated to the national level, these 'unnecessary treatments' represent a potential loss of approximately $339,000 per year (95% CI $47,000-$1,149,000), a substantial loss in a country which in 2016 had an estimated total health care expenditure (out of pocket, government and donor) of approximately $4 billion based on a per-capita expenditure of approximately $82 per capita (at 2018 purchasing power parity) [42].

The basic cost-effectiveness model utilised in this manuscript does not incorporate the opportunity costs to the patients of testing, treatment or re-visits after unsuccessful treatments, as we do not have sufficient data to accurately estimate these. However, they are important aspects, and their omission indicate that our model over-estimates the cost-effectiveness of the FBAT from a societal perspective. The model also assumes a 100% compliance with treatment (i.e. all patients diagnosed and provided with a prescription go ahead and acquire the prescribed drugs). While all participating clinicians confirmed that they issue a prescription to all patients testing positive to the FBAT, they could not provide us with an estimate of how many patients go on to buy the treatment as this is likely to depend on many other factors. Therefore, while we recognize that this assumption is unlikely to be true and may lead to an overestimation of the cost-effectiveness results, we currently have no data under which to make a probability distribution for the treatment compliance. We therefore feel it is more appropriate to assume 100% for each diagnostic test scenario. We do acknowledge, however, that improvements in diagnostic performance may influence patient trust and there may be a differential compliance effect for the two tests, which again we are unable to parameterise at present. We suggest that the influence of diagnostic test performance on clinician prescription decisions and patient compliance would be prime territory for additional research within the resource constrained communities we work within. Furthermore, the national-level extrapolation of economic losses due to misdiagnosis should be interpreted with caution due to the use of a single national-level prevalence estimate which was based upon the 2007 Kenya AIDS Indicator Survey, which also highlighted the very variable regional estimates of brucellosis prevalence [19], rather than prevalence within febrile individuals seeking health care and tested for brucellosis based on clinical judgement. A more robust analysis should next be undertaken utilising regional-specific data on the prevalence and number of individuals tested to increase the accuracy of this estimate.

## Study limitations

We recognize that the relatively limited size of the sample set, together with the small number of positive individuals and consequent wide confidence intervals on many estimates, might limit the external validity of the study. Furthermore, we were unable to carry out a second test to determine the diagnostic performance of the FBAT and RBT in this study. Finally, we acknowledge that some of the assumptions, particularly those regarding patient treatment compliance, might have influenced our cost-effectiveness results. However, the main scope of the study was to determine the cost-effectiveness of the test while keeping all other parameters (e.g. patient compliance) constant.

## Conclusion

Our recommendation, based on the evidence in this study, is that health authorities in Kenya and elsewhere should rapidly move away from dependence on the FBAT, primarily for performance and cost reasons, but also for ethical reasons viz a viz providing appropriate diagnostic results to patients and for safeguarding of the therapeutic efficacy of agents used to treat brucellosis (primarily antibiotics). The RBT using a high-quality antigen from a trusted source

should be considered as the nationally approved diagnostic tool for point-care decision-making on brucellosis as it is simple to run and relatively inexpensive. Suspicious cases should then be confirmed using a secondary serological test, such as the Immunocapture Agglutination Test [5,22]. We also recommend that the treatment regimens are harmonized to allow for an appropriate and judicious use of antibiotic drugs. In recent years, hospitals in Kenya have embraced the Standard Agglutination Test for *Salmonella typhi* O and H antibody titres, instead of the previously used Widal test, for diagnosis of salmonellosis. We are therefore confident that healthcare professionals will be willing to change to more specific diagnostics for brucellosis as they realize that the currently used tool is causing more harm because of over-diagnosis and misdiagnosis. Furthermore, endorsement from the national government will allow for large-scale procurement of RBT reagents, further improving the cost-effectiveness of RBT while promoting its use among hospitals nation-wide.

## Acknowledgments

The authors thank the participating hospitals for their ongoing support and compliance, and Prof. Ian Dohoo for advice on data analysis.

## Author Contributions

**Conceptualization:** Lorren Alumasa, Fredrick Amanya, Samuel M. Njoroge, Eric M. Fèvre, Laura C. Falzon.

**Data curation:** Lian F. Thomas, Samuel M. Njoroge, Laura C. Falzon.

**Formal analysis:** Lian F. Thomas, Laura C. Falzon.

**Funding acquisition:** Eric M. Fèvre.

**Investigation:** Lorren Alumasa, Fredrick Amanya, Samuel M. Njoroge, Josiah Makhandia.

**Methodology:** Lorren Alumasa, Lian F. Thomas, Fredrick Amanya, Samuel M. Njoroge, Ignacio Moriyón, Josiah Makhandia, Jonathan Rushton, Eric M. Fèvre, Laura C. Falzon.

**Project administration:** Eric M. Fèvre, Laura C. Falzon.

**Resources:** Ignacio Moriyón, Eric M. Fèvre.

**Software:** Lian F. Thomas, Laura C. Falzon.

**Supervision:** Eric M. Fèvre, Laura C. Falzon.

**Validation:** Lian F. Thomas, Jonathan Rushton, Eric M. Fèvre, Laura C. Falzon.

**Visualization:** Lian F. Thomas, Eric M. Fèvre, Laura C. Falzon.

**Writing – original draft:** Lian F. Thomas, Laura C. Falzon.

**Writing – review & editing:** Lorren Alumasa, Lian F. Thomas, Fredrick Amanya, Samuel M. Njoroge, Ignacio Moriyón, Josiah Makhandia, Jonathan Rushton, Eric M. Fèvre, Laura C. Falzon.

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
