## [Decision Letter · Decision Letter 0]

9 Jul 2020

Dear Dr. Laura Cristina Falzon,

Thank you very much for submitting your manuscript "Hospital-based evidence on brucellosis diagnosis and treatment in Kenyan hospitals" for consideration at PLOS Neglected Tropical Diseases. As with all papers reviewed by the journal, your manuscript was reviewed by members of the editorial board and by several independent reviewers. In light of the reviews (below this email), we would like to invite the resubmission of a significantly-revised version that takes into account the reviewers' comments. 

We cannot make any decision about publication until we have seen the revised manuscript and your response to the reviewers' comments. Your revised manuscript is also likely to be sent to reviewers for further evaluation.

Sincerely,

Vasantha kumari Neela

Associate Editor

Marco Coral-Almeida

Deputy Editor

Reviewer's Responses to Questions

**Key Review Criteria Required for Acceptance?**

**Methods**

-Are the objectives of the study clearly articulated with a clear testable hypothesis stated?

-Is the study design appropriate to address the stated objectives?

-Is the population clearly described and appropriate for the hypothesis being tested?

-Is the sample size sufficient to ensure adequate power to address the hypothesis being tested?

-Were correct statistical analysis used to support conclusions?

-Are there concerns about ethical or regulatory requirements being met?

Reviewer #1: Authors use data collected from hospitals in Kenya and results obtained with the same patient sera in both tests to calculate their cost-effectiveness, from a societal perspective, defined as the cost in US Dollars ($) per Disability Adjusted Life Year (DALY) averted over the course of one year. The objective of the work is clear and the design of the study, sample size and statistical analysis seem appropriate to address the stated objectives. According to the manuscript, ethical and regulatory requirements were met. However, there are several gaps and inconsistencies in the manuscript that need to be clarified and properly described before considering the work for publication:

1. The major gap of the article is the lack of a precise description of the sera used for comparison. In the abstract it is stated that “serum samples that were tested for brucellosis using the FBAT, were later re-tested using the RBT”. It is also indicated: “FBAT results were available for 205 patients, of which 30 (15%) tested positive”. Accordingly, these 205 serum samples should have been re-tested with the RBT. Surprisingly, it is indicated (abstract and lines 263-264): “A total of 255 serum samples were collected from the 10 hospitals, and these were re-tested with the RBT”. Thus, results in Table 2 are confusing since the number of sera tested in each test is different and the corresponding number of positive/negative is not indicated. In addition to an error in the percentage of FBAT positive samples in hospital A (should be 50% instead of 5%), other source of confusion comes from data in hospital J since it is impossible to assess if the only RBT positive sample identified was previously positive or negative in the FBAT, and if the 2 FBAT positive samples were negative in the RBT. The correspondence of FBAT/RBT results cannot be assessed also in the other hospitals. If the objective is to show the correspondence among FBAT and RBT results this Table should be fully amended indicating clearly which FBAT positive/negative sera were also positive/negative in RBT. However, authors should consider if this table could be unnecessary and redundant with data in Table 3. To increase the mess, authors indicate (lines 265-266): “Both FBAT and RBT results were available for 180 samples (Table 3)”, and the relative DSe/DSp and relative (to be amended along the manuscript) False Positive/Negative values calculated. Altogether, the precise number of sera tested should be well defined in the material and methods, and the corresponding tables amended/merged.

2. In addition to the precise description of the sera used, the material and methods section should be properly completed with a description of the FBAT test (origin, antigen composition, method, etc…). To have a better (buy yet incomplete) idea on FBAT methodology is necessary to read lines 289-293 (i.e., the results section). Moreover, comments in lines 256-261 should be explained better. It is stated that four hospitals conducted FBAT with both A and M antigens. Which FBAT antigen (A?, M?) was used in the remaining hospitals? It would be probably more practical avoiding all these comments (based in inadequately assumed test properties) in the results section, and make suitable criticisms of FBAT in the discussion section (that in fact were made yet -lines 346/347-).

3. Table 3. It is indicated that Scenario 2 was built according “Brucella spp. prevalence in febrile patients presenting to hospital in North-East Kenya based on real time PCR”. This seems to be out of the context of the study. If this test is to be included in the comparative study, a precise description of the PCR protocol should be made in the material and methods section, and moreover, the correspondence with the serological results properly described also. Otherwise this scenario should be omitted from the manuscript.

Reviewer #2: The study design is appropriate to address the stated objectives of seeking hospital-based evidence on brucellosis diagnosis.

Reviewer #3: -Are the objectives of the study clearly articulated with a clear testable hypothesis stated?

The statement of aims and objectives could be clearer. I’d suggest replacing the final introduction paragraph which covers the background to the study with a clearer statement of study objectives. 

-Is the study design appropriate to address the stated objectives?

Yes – 

Can you clarify if you collect any individual level data on the actual treatments prescribed for the individuals whose samples were tested? This is not really clear from the methods. If the only data gathered were the ‘recommended’ treatment regimen for brucellosis at hospital level then more discussion on the limitations of this approach is needed. The recommended treatment regimen is likely to be a poor predictor of actual practice which will vary by clinician and all clinicians are likely to put < 100% FBAT positive individuals on brucellosis treatment. Your estimates of costs due to FBAT will therefore be inflated. 

As you compare the FBAT to the RBT – some content in the introduction on the known limitations of the RBT would be helpful to prime the reader to better interpret the data presented. The implications for the interpretations of the study findings are not large – but should be made clear throughout. Content on this is given in the discussion but could be moved to the introduction instead. 

-Is the population clearly described and appropriate for the hypothesis being tested?

Can you clarify the ‘inclusion criteria’. In a few places you talk about the implications of these findings for inferences about febrile populations but I think these data are specific to individuals who are tested for brucellosis based on clinical judgement? While these populations are likely to overlap not all individuals tested for brucellosis will be febrile and <100% febrile presentations are likely to be tested for brucellosis? 

-Is the sample size sufficient to ensure adequate power to address the hypothesis being tested?

There is no mention of power. Given the relatively limited size of the sample set, small number of positive individuals and wide CI on many estimates this should be explicitly acknowledged. 

-Were correct statistical analysis used to support conclusions?

In specifying the betapert parameters can you clarify the rationale for the selection of the max and min values specified. It looks like you have used the 95CI values (e.g. for p4) which perhaps sets the distribution quite conservatively narrow (e.g. using 5% and 95% as 0 and 100% values)? Is this standard? 

Also – please provide more information on which FBAT the literature estimates used to parameterise this model are based on (from Kiambi et al study) and how this relates to the tests used to generate the data in this study? 

Data that I am aware of (admittedly unpublished) would indicate that you’d need a much wider CI on this value to capture e.g. the range of performance of 4 FBAT assays (~ 10-80%) so I would be interested in the implications of a less precisely specified parameter for the findings of this analysis. 

P13 is based on very limited data I believe (n=2 observations) . Allowing a little more variation in this parameter would seem sensible – not least because, as you indicate costs might well be lower at scale. 

P15 – how do you specify a 3 parameter distribution with the 2 values given? 

Line 233-234 -can you add more detail on the methodology applied for a ‘basic sensitivity analysis’? 

It is important that you include more details on the actual FBAT kits used – e.g. manufacturer and product names. There are a variety of options in the setting I am familiar with and their performance is quite variable. I think it is important to make this clear and move away from the impression that you give that the FBAT is a single fixed test. 

In Table 1 – can you clarify which FBAT the estimated for Params 4-7 were estimated using? Are these appropriate for all of the tests used for this study? 

I appreciate the importance of the comparison of the relative performance of the FBAT and RBT but the absence of any additional confirmatory testing of these samples is a limitation that should be recognised. It is also important that you clarify the need (according to all of the guidelines that I am aware of) for a confirmatory serological test. Confirmatory testing may not be performed in practice for many individuals but should be mentioned before the section in the discussion which indicate that secondary tests are as an option. 

-Are there concerns about ethical or regulatory requirements being met?

In the methods (line 137) you refer to extracting data on patient ID but there is apparently no consent obtained from participants. Why not? Please give some additional explanation of the handling of personal identifying info and why consent was not obtained.

Reviewer #4: Introduction

1. Please tighten the write up, keeping it crisp and relevant to the topic. 

2. L112-119 The quote from the worker is not necessary. Please remove. 

3. Please make objective (s) of the study explicit and clearer. In my opinion, there are several objectives in this study. Hence, those need to be explicitly stated but at the same time, relevant to each other.

Materials and Methods

1. Was permission requested from the hospital management and administrator or Ministry of Health to conduct the study? If it was, please incorporate. The way it is written suggested that only the clinician or lab technologist were communicated. 

2. Please include ethical clearance ID (or other identification) from the board. 

3. Please include description on why those hospitals were selected, it is not clear if these are all available hospitals in the study regions. Include criteria for selection if any. 

4. Please arrange and flow the materials and methods according to the objectives statement.

5. Please make study design and sample size determination clearer.

6. It is not clear what Levels of the hospital mean, is the relevant to the design of the study. Please clarify or remove.

7. Please be clear about the published Se and Sp of each tests. Since the inaccuracy of FBAT was noted several times by the author and forms the backbone of the study, the Se and Sp should be made explicit. It would also be useful to note how the cited authors determined that FBAT diagnostic ability is poor. 

8. In my understanding, the authors attempted to determine the diagnostic accuracy of FBAT by comparing its performance to RBPT which is not a gold standard reference. Although the Se and Sp of the two imperfect tests were not stated, the authors stated that RBPT is the more accurate test therefore used the findings from the RBPT test to calculate the ‘relative’ Se and Sp of FBAT. At the same time the agreement was also calculated. Since the Se and Sp were not stated and there was no attempt by the authors to determine the true positive and true negative of RBPT before comparison to FBAT was performed, there could be questions about misclassification bias when RBPT diagnosis was then used to compare to FBAT and these were then used as input for the mathematical models. 

9. Please also look into using the method suggested by Albert (Estimating diagnostic accuracy of multiple binary tests with an imperfect reference standard. Stat Med, 2009) to determine the diagnostic accuracy of FBAT. 

10. Please be clear on what is meant by ‘societal perspective’ for cost effectiveness. 

11. L185 – L193 is very confusing, please rectify.

12. L244 – records cannot be tested for brucellosis. Please revise. 

13. Please revise comparison of the two tests considering my above comments on Se and Sp. 

14. It was not clear if both RBPT and FBAT were performed by the authors or if only RBPT was performed and results were compared to FBAT results performed by the hospitals.

**Results**

-Does the analysis presented match the analysis plan?

-Are the results clearly and completely presented?

-Are the figures (Tables, Images) of sufficient quality for clarity?

Reviewer #1: Main concerns on results and discussion are:

4. Table 4. It is remarkable (and incomprehensible to this reviewer) why the same treatment is 5 times more expensive in hospital H than in hospital B. If this is not an error, this finding should be better commented and discussed properly.

5. Lines 357-358: that the “prozone” is the only (or the major) responsible of the poor diagnostic performance of FBAT is very questionable. Improper setting up (antigen composition and preparation) and validation, among other problems, can explain also the poor FBAT performance.

6. Lines 463-465. The consideration regarding the suboptimal specificity of the RBT is not correct and I suggest a careful revision of this sentence. The great sensitivity of RBT to detect Brucella specific antigenic stimulus in endemic settings can never be considered as a lack of specificity. 

7. Lines 475-476: In brucellosis, the concept of “screening” and “confirmatory” tests is ambiguous and frequently misinterpreted. In animal brucellosis (when vaccination is used), a serial RBT (as “screening”)/CFT (as “confirmatory”) testing can be of value to minimize the number of unnecessarily culled animals. However, in human brucellosis the RBT is not a “screening” test that, in opposition to other diagnostic tests, needs to be “confirmed”. If properly performed the RBT is specific enough and results in the same diagnostic performance as, for example, SAT, ELISA and LFIC. Importantly, the results of any indirect diagnostic test have to be contrasted by the clinician with the epidemiological and clinical evidence and, if available, cultures. I strongly suggest authors to discuss this issue with enough care to avoid perpetuate this misconception.

Reviewer #2: The results are clearly and completed presented.

The two legends in Figures 4 and 5, Input High and Input Low, are difficult for the readers to distinguish.

Reviewer #3: -Are the results clearly and completely presented?

Can you give the raw data on test results so that the breakdown by FBAT manufacturer and hospital is clear. You have the hospital breakdown in Table 2 – was the same test used for all tests at a given hospital? Can you add the detail on the tests used? The FBAT is not a single test so this needs to be clear in the presentation of the data and in the discussion of the results.

You state a series of assumptions (pg 11) but don’t really come back to many of these in the discussion. How plausible is to that each patient treated undergoes a full brucellosis treatment protocol? Or that patients seek care at the onset of symptoms? 

Given widespread awareness of the limitations of brucellosis testing (amongst patients and clinicians) might assumption 7 not change with improved diagnostics? 

What are the implications of violating these assumptions for the interpretation of your findings?

You recognise the assumption that all individuals who get a positive test result are treated for brucellosis. In my experience only a fraction of individuals receiving an FBAT positive result are put on brucellosis consistent treatment regimens. As you recognise in the paper this is largely because clinicians are well aware of the limitations of these tests. Can you include a parameter for % test positive individuals treated for brucellosis? I’d imagine this will vary with both prevalence and PPV of the test used. 

Line 244-and table 2

As the bulk of your analysis depends on data where both the FBAT and RBT were performed (n=180), the data presented in table 2 could be made much clearer. If “Serum samples available” = “n tested by RBT” can you make this clearer. Can you give clearer labels to the 2 columns currently headed “% positive” to differentiate the tests

Can you also clarify which if any estimates presented or used in the model use a different denominator – ideally stating the population clearly in each case. 

Lines 256-261 – can you give a more systematic presentation of these data (and the methods followed to generate them)? Who did you ask about test info? What were the manufacturers of the tests? Which hospital was each test used in? What data do you have to support the statement that ‘consequently a prescription was provided by clinicians’. Did you ask explicitly if all test positive individuals were put on brucellosis treatments? How plausible do you think that is? 

Much of the data on test results for abortus and melitensis antigens is not key and presenting them risks giving these tests a validity that is not warranted. Rather than presenting these results here you could instead include a clearer statement of the definitions of ‘test’ positive by FBAT that should be given in the methods. E.g. for FBAT, individual classified as positive if any reaction at any dilution with any of the antigens in the specific kit? The details of the actual kits used are needed here and any information gathered on the way they are applied. Also – presumably FBAT positive means reported positive by any protocol reported – with or without dilutions etc? This is important detail to give to enable your reader to determine the validity of comparing the FBAT and RBT, and handling of the FBAT as a single test. 

Line 263 – value of RBT testing sera for which no FBAT data are available is not clear? Why was this done? 

There are results presented in this section that aren’t really introduced in the methodology. Some more detail on the respondents per hospital and the exact questions asked to obtain the data presented would be helpful. 

Lines 288-294 – You have mentioned the protocol options given in test kit inserts but not the details of the protocols actually run at the hospitals. Did any run dilutions - slide/ or tube format? If so, did you specify a common protocol for the question about time to perform one test? 

You mention the publication dates of the references in the test protocols but make no point about this in the discussion – presumably the implication is that these are outdated kits/technology? 

Line 294 Can you clarify the question asked to obtain the ‘mean cost of a test/patient’ in the methods and indicate clearly if this is a cost per test or per patient here. 

Line 300 – given that treatment regimens are either consistent with guidelines or not and that durations will vary by drug I am not sure how meaningful a mean treatment duration is? 

Line 308 – language point – I think the shift to RBT would be cost-effective or not as compared to the FBAT? Vs more cost-effective.

Lines 325-328 – it would be helpful to spell out the directions of the relationships for the most influential parameters – e.g. lower prevalence > what implications for cost differences? In what contexts would cost differential be most pronounced? You mention this in the discussion (lines 458-460) but it takes a long time to get to. 

-Are the figures (Tables, Images) of sufficient quality for clarity?

Table 2 – 

Can you include the kit name/manufacturer info for the FBATs used. 

Table 4 – 

Please give more detail in the methods on the questions asked about the “protocol of the FBAT”. Was this “how long does it typically take to run the test?” or other details also? Is the ‘test cost’ the charge to the patient or another quantity? 

Please include rows for all hospitals with NAs indicating missing data as needed? 

Can you add a column to indicate which of these treatments regimens are consistent with WHO guidelines? 

Figures – can you plot the distributions of the $/DALY averted estimates for each scenario – not just for scenario 1 where prevalence is very low and include the plots on a single figure to aid comparison (e.g. table 5 as graphic)

Can you include more detail in the legends for figures 4 and 5 – e.g. what is “DW”? And other abbreviations used

In the discussion of the $/year for treatment of false positives (re results in Table 5 and Lines 318-324) it would be useful for you to compare the costs for brucellosis treatment to the costs for treatment for other causes of febrile illness. Given that the patients currently testing positive by FBAT would be treated for something if not brucellosis and that this would presumably have some cost also are the values for $/year treating false positives not overestimates?

Reviewer #4: 1. Please align results to the objectives and materials and methods.

2. Please improve how results are presented, but keeping them concise and structured. At present, the results are hard to follow and since the objectives are not clear, it is hard to engage the results to a stipulated objective. 

3. There are too many tables and figures in this manuscript, please consider merging some and deleting others that has been described in text. Some of the figures are hard to read, please improve quality.

**Conclusions**

-Are the conclusions supported by the data presented?

-Are the limitations of analysis clearly described?

-Do the authors discuss how these data can be helpful to advance our understanding of the topic under study?

-Is public health relevance addressed?

Reviewer #1: Conclusions of the study are right and provide an useful support to hospitals to address a public health relevant problem (brucellosis), particularly in scarce resource areas.

Reviewer #2: (No Response)

Reviewer #3: -Are the limitations of analysis clearly described?

Not all – Some more explicit discussion of the fact that the estimates for cost of treatment based on FBAT are likely overestimates as not all individuals with positive test results will be put on brucellosis treatment is needed. 

-Do the authors discuss how these data can be helpful to advance our understanding of the topic under study?

-Is public health relevance addressed?

Yes – this content is covered.

Lines 339-341 might be a bit premature. While this statement is hopefully true, ‘will’ might not be the best term. I also think it important to recognise the limited scale of this study in the discussion.

Reviewer #4: Discussion

1. Much of the discussion needs to be revisited after taking account the diagnostic accuracy of FBAT.

2. Please improve conciseness, reduce wordiness and flow throughout.

3. L349-353 – this is speculation, unless there is type of suggestive evidence please remove.

4. Include limitations of the study, including limitation of the test method used, mathematical modelling and data collections methods. 

5. Please tie conclusion to objectives of the study.

**Editorial and Data Presentation Modifications?**

Reviewer #1: Minor corrections:

- Line 70. Authors should not use clinical/epidemiological adjectives such as “chronic” brucellosis to define the clinical course of the disease. From the scientific standpoint it would be probably better use terms like “short/long evolution”. 

- Lines 110-119. I strongly suggest deleting this personal (literal) opinion of the officer, and to be replaced with a more clear and suitable comment to illustrate the issue. 

- Line 178. Delete “from a societal perspective” since it is already indicated in line 176

- Line 185. To avoid difficulties to readers, I suggest introducing “The different parameters and scenarios for the comparative cost-efficacy analysis are shown in Table 1” or a similar sentence.

- Table 1. Unless this be a rule of the journal, the meaning of P (probability) is missing. It should be indicated in the text (line 180. “associated probability (P)”) and in Table 1 (as foot note).

- Line 202. Replace “patents” by “patients”

- Figure 1. The meaning of P, Se, Sp, Dx and Tx is missing (it should be indicated in the figure caption). The abbreviations Se and Sp should be included also in the text (in brackets) the first time these terms appear. 

- Line 320. Replace “under an RBT” by “under a RBT”

Reviewer #2: Page 18, there is a typo for the line ‘Hospital A’ in Table 2. For FBAT, % positive should be 3/6=50%, not 5%.

Reviewer #3: Line 40 and 65 – a single test is never sufficient to make a confirmatory diagnosis of brucellosis either. The inclusion of two tests in international guidelines should be made clear here. 

Line 83 – ‘sensitivity/specificity balance’ is quite vague. Can you clarify which test performance metrics are dependent what epi conditions? 

Line 90-91 – you imply that many studies from markedly different settings have all shed light on some common characteristics – or indicate endemicity – but this is not very clear. Can you clarify

Line 101 – given that you have described brucellosis as endemic in Kenya is re-emergence the right term -or can you provide more precision on the contexts referred to? 

Lines 120-126 – the narrative on the origins of the study is not really key to this paper. Can you give more info on the scientific rationale that presumably motivated some of the interest of stakeholders?

Line 142 – does the ‘diagnostic kit protocol’ mean manufacturer info and specifications on test performance? Could be clearer. 

Lines 160 – the reference to paper based records is not very clear – what records are these? 

Line 188 – how realistic is this assumption that all parameters apart from prevalence would remain the same? See suggestion above about including additional parameter for % FBAT positive individuals put on treatment, which might be expected to depend on PPV, which in turn depends on prevalence. 

Line 354- The prevalence estimates and other test performance metrics reported here are somewhat misleading as they apply to different populations – can you report the % positive and other stats obtained through analysis of the population tested by both tests (e.g. ~13.3% vs 1.66 %? for prevalence)

Line 358 – do you know that the FBAT kits are all pH neutral? What data is this based on? Can you include a reference? 

Line 380 – really ? if malaria is likely to be tested for alongside brucellosis how likely is it (given the availability of high quality rapid tests for malaria, relative severity of the diseases and emphasis on malaria in clinical training as well as societal pressures not to miss malaria cases) that any/many cases of malaria would be missed and treated as brucellosis? 

Lines 383 -the points about common risk factors for multiple zoonoses are not very clear here. Can you explain further?

Line 400 – can you add the data on compliance with WHO recommendations to Table 4 vs reporting results in the discussion

Line 409 – can you clarify what practice you are defining as ‘improper’ use of drugs in this paragraph? 

Lines 421 – the points you make here are entirely consistent with a scenario in which clinicians interpret the positive FBAT alongside their knowledge of the poor performance and low positive predictive value of these tests and other data and choose to put patients on a short course of antibiotics to try to treat multiple possible bacterial infections. In contexts that I have worked in clinicians given an initial treatment like short course of doxycycline and then only consider shifting to a brucellosis drug regimen if patients fail to respond to that treatment. What are the implications of such practices for your assumptions re treatment of all FBAT positives for brucellosis? 

Line 430 – what ‘other febrile issues’ do you mean? 

Line 434 – can you present the multiplier given here (9.2) explicitly in the results - with CI also? 

Line 439 – can you give a date for the $197 purchasing power parity figure – the language here is a little unclear on the comparability of these figures. 

Line 443 – I am confident that the RBT is cost effective vs the FBAT but if all FBAT positive individuals were put on brucellosis treatments that would likely be effective treatment of most bacterial infections that might have caused their illness – leading to relatively low rates of treatment failure and repeat visits perhaps? However, you have likely over-estimated the costs of the FBAT with this assumption so a bit more reflection on these assumption would be sensible.

Lines 446 – you could also mention the very variable regional estimates of brucellosis prevalence presented in this reference.

Lines 447 – See earlier point on the distinction between febrile individuals and individuals tested for brucellosis. Data on these proportions and the factors that determine testing practices (which are likely to be influenced by test performance) would be valuable in performing a next step analysis of cost-effectiveness also. 

Line 455 – the focus on neurobrucellosis is a little strange here. What proportion of brucellosis presentations have neurological signs in this context? How relevant is this focus as compared to say joint manifestations? How many facilities are equipped for safe testing of CSF? 

Line 477 – given the existence of abundant existing data on the superior performance of the RBT vs FBAT tests what other factors do you think might be important in achieving a change in testing behaviour? Is ‘realisation’ of the poor performance of FBATs the only factor?

Reviewer #4: (No Response)

**Summary and General Comments**

Reviewer #1: The objective of the work submitted is to assess the comparative performance and cost-effectiveness of the Rose Bengal (RBT) and Febrile Antigen Brucella Agglutination (FBAT) tests for diagnosing human brucellosis to support hospitals in Kenya in creating evidence regarding the most appropriate diagnostic strategy. Overall, the study is well focused and the manuscript well written. The study highlights the misdiagnosing problems generated by FBAT and points out the RBT as mainstay diagnostic test for human brucellosis. Moreover, authors also identify the need for harmonization of treatment guidelines in Kenyan hospitals. Conclusions achieved add further evidence to the usefulness of RBT, an often neglected and misunderstood test, particularly in brucellosis endemic areas of resource limited countries. Hence, this work is worth to be published to support hospital decision makers in resource limited settings to implement appropriate diagnostic methods and treatments. However, several gaps and important inconsistencies have been identified (see methods and results sections), and therefore, to be acceptable for publication the manuscript should be submitted to a careful review. To this end, my comments are made in the hope that they will be well-taken and useful to the authors.

Reviewer #2: Regarding the diagnosis of brucellosis in Kenya, the authors compared the diagnostic performance and cost-effectiveness of the routinely used FBAT with that of the Rose Bengal Test (RBT) , and provided the evidence that the RBT was a more cost-effective diagnostic test. 

RBT has been widely adopted in many countries, from the economic perspective the authors conducted the cost-effectiveness in detail to indicate the superiority of RBT, while there must be several factors that affect the application of RBT in Kenya. The authors should add some discussion about these factors, and only by addressing these factors, the application of RBT can be promoted.

Reviewer #3: This paper presents useful data and analyses on an important topic with considerable public health and policy relevance. The study has some limitations but makes a useful contribution to the literature. Some modifications are needed. 

Key points that apply throughout are the need to clarify that 1) these data come from multiple different FBATs not one single test and 2) the problems with the assumptions in the cost calculations that all FBAT positive individuals would be put on the recommended treatment for brucellosis. I can see the need to make these assumptions but if more precision in the estimation is not possible (see some suggestions in comments), then these limitations do at least need to be more explicitly recognised.

The balance of content across sections also needs review. There is much more content in the results and discussion sections on antimicrobial treatment regimens than indicated based on the introduction and methods and some of the results presented have limited detail and set up in the methods sections.

Reviewer #4: I appreciate the work of Alumasa et al and in my opinion this work presents several important information that can be useful in countries where Brucellosis is endemic. According to the authors, the FBAT test is widely used in many countries in Africa including Kenya for the diagnosis of Brucellosis even though it is unspecific and costly, therefore has resulted in over diagnosing and unnecessary treatments in hospitals. The work attempted to determine the type and cost of treatment for Brucellosis through the use of mathematical modelling to simulate cost outcomes based on the various presented scenarios. However, I am concern about the vagueness of the method of determining the diagnostic accuracy of FBAT. I have included some suggestions for the authors to follow through. I am also concern about the length and wordiness of the manuscript. In my opinion the paper can be made significantly more concise and less verbose for the convenience of the reader. I recommend professional editing to improve the flow and readability of the manuscript. Please improve the title of this manuscript as it is not reflecting the study scope and content.

PLOS authors have the option to publish the peer review history of their article (what does this mean?). If published, this will include your full peer review and any attached files.

Reviewer #1: No

Reviewer #2: No

Reviewer #3: No

Reviewer #4: No
---

## [Decision Letter · Decision Letter 1]

10 Nov 2020

Dear Laura Cristina Falzon,

We are pleased to inform you that your manuscript 'Hospital-based evidence on cost-effectiveness of brucellosis diagnostic tests and treatment in Kenyan hospitals' has been provisionally accepted for publication in PLOS Neglected Tropical Diseases.

Please complete the minor corrections commented by the reviewers.

Best regards,

Vasantha kumari Neela

Associate Editor

Marco Coral-Almeida

Deputy Editor

Reviewer's Responses to Questions

**Key Review Criteria Required for Acceptance?**

**Methods**

-Are the objectives of the study clearly articulated with a clear testable hypothesis stated?

-Is the study design appropriate to address the stated objectives?

-Is the population clearly described and appropriate for the hypothesis being tested?

-Is the sample size sufficient to ensure adequate power to address the hypothesis being tested?

-Were correct statistical analysis used to support conclusions?

-Are there concerns about ethical or regulatory requirements being met?

Reviewer #1: (No Response)

Reviewer #4: The authors have made significant and important changes as requested by the reviewers. This paper is now more concise and clearer.

**Results**

-Does the analysis presented match the analysis plan?

-Are the results clearly and completely presented?

-Are the figures (Tables, Images) of sufficient quality for clarity?

Reviewer #1: (No Response)

Reviewer #4: Yes.

**Conclusions**

-Are the conclusions supported by the data presented?

-Are the limitations of analysis clearly described?

-Do the authors discuss how these data can be helpful to advance our understanding of the topic under study?

-Is public health relevance addressed?

Reviewer #1: (No Response)

Reviewer #4: Yes.

**Editorial and Data Presentation Modifications?**

Reviewer #1: Minor corrections:

Title: according to the authors the title has been revised but there are no differences between the original and the revised title (please check)

Line 23: I suggest “180 patient serum samples” instead of “180 serum samples”

Line 65: I suggest “differential diagnosis” instead of “confirmatory diagnosis”

Reviewer #4: Minor comments:

1. Whenever available, please state the Se and Sp for FBAT. Also please state the published Se and Sp for RBPT.

2. Table 2 first line need some explanation on the 'NA'.

**Summary and General Comments**

Reviewer #1: The authors of the manuscript have made a great effort to undertake all the modifications suggested by the 4 reviewers. The work has improved considerably and it is definitely recommended for publication. I have just few minor corrections to suggest.

Reviewer #4: This paper provides the evidence for a data-driven decision and policy for healthcare sector in Kenya that will benefit the country by reducing cost and minimizing unnecessary therapy for a very important endemic zoonoses.

PLOS authors have the option to publish the peer review history of their article (what does this mean?). If published, this will include your full peer review and any attached files.

Reviewer #1: No

Reviewer #4: No

---

## [Editor Report · Acceptance letter]

30 Dec 2020

Dear Dr. Falzon,

We are delighted to inform you that your manuscript, "Hospital-based evidence on cost-effectiveness of brucellosis diagnostic tests and treatment in Kenyan hospitals," has been formally accepted for publication in PLOS Neglected Tropical Diseases.

Best regards,

Shaden Kamhawi

co-Editor-in-Chief

Paul Brindley

co-Editor-in-Chief
